# Metaheuristics in the Humanitarian Supply Chain

Francisca Santana Robles [1], Eva Selene Hernández-Gress [2,*], Neil Hernández-Gress [2] and Rafael Granillo Macias [1]

[1] Higher Education School Ciudad Sahagun, Autonomous University of the State of Hidalgo, Pachuca 43990, Hidalgo, Mexico; profe_7739@uaeh.edu.mx (F.S.R.); rafaelgm@uaeh.edu.mx (R.G.M.)
[2] Tecnologico de Monterrey, Pachuca 42080, Hidalgo, Mexico; ngress@tec.mx
* Correspondence: evahgress@tec.mx

**Abstract:** Everyday there are more disasters that require Humanitarian Supply Chain (HSC) attention; generally these problems are difficult to solve in reasonable computational time and metaheuristics (MHs) are the indicated solution algorithms. To our knowledge, there has not been a review article on MHs applied to HSC. In this work, 78 articles were extracted from 2016 publications using systematic literature review methodology and were analyzed to answer two research questions: (1) How are the HSC problems that have been solved from Metaheuristics classified? (2) What is the gap found to accomplish future research in Metaheuristics in HSC? After classifying them into deterministic (52.56%) and non-deterministic (47.44%) problems; post-disaster (51.28%), pre-disaster (14.10%) and integrated (34.62%); facility location (41.03%), distribution (71.79%), inventory (11.54%) and mass evacuation (10.26%); single (46.15%) and multiple objective functions (53.85%), single (76.92%) and multiple (23.07%) period; and the type of Metaheuristic: Metaphor (71.79%) with genetic algorithms and particle swarm optimization as the most used; and non-metaphor based (28.20%), in which search algorithms are mostly used; it is concluded that, to consider the uncertainty of the real context, future research should be done in non-deterministic and multi-period problems that integrate pre- and post-disaster stages, that increasingly include problems such as inventory and mass evacuation and in which new multi-objective MHs are tested.

**Keywords:** disasters; humanitarian supply chain; metaheuristics

## 1. Introduction

The Humanitarian Supply Chain (HSC hereafter) has become a major research topic since the 2004 Indian Ocean tsunami that blocked airports in affected areas. Water, food, clothing, medicine and basic necessities, among other things, must be supplied to victims of both natural and man-made disasters. In addition to this, pandemics make it necessary for medicines and vaccines to be distributed to the general population, so doing it in the fastest way, with greater coverage to the population, and at the lowest cost, is an issue that is attracting attention of researchers and practitioners.

Natural disasters might be earthquakes, hurricanes, floods or droughts, among other things. As can be observed in Figure 1, even though from 2000 to 2020 the number of disasters remained constant, from 1960 to this date they have been increasing. Moreover, some disasters in different areas have been more significant, for example, the earthquakes in Japan and some Latin American countries like Mexico.

Figure 2 shows the countries with the highest number of natural disasters in 2020, with Indonesia having the most, with 29, followed by the United States with 23.

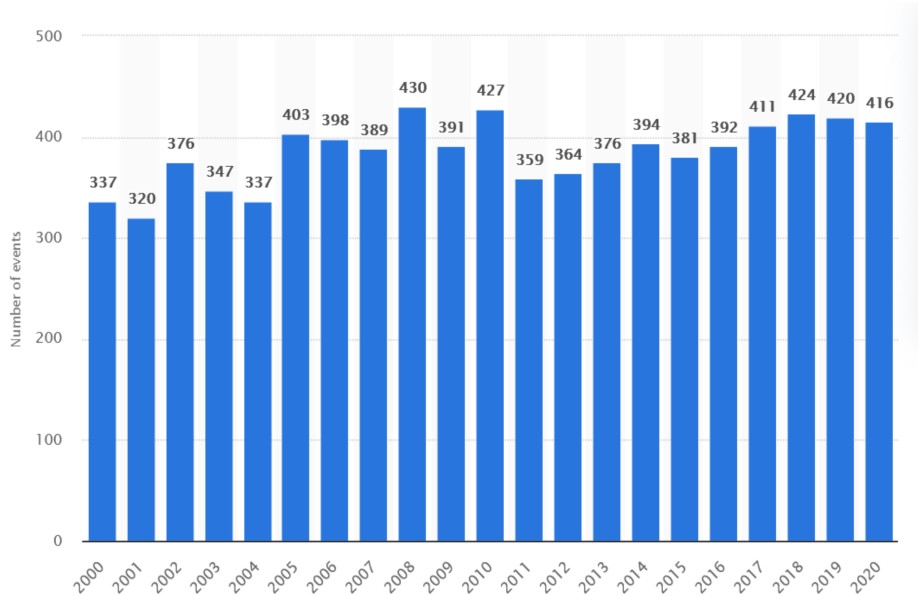

**Figure 1.** The annual number of natural disaster events globally from 2000 to 2020 [1].

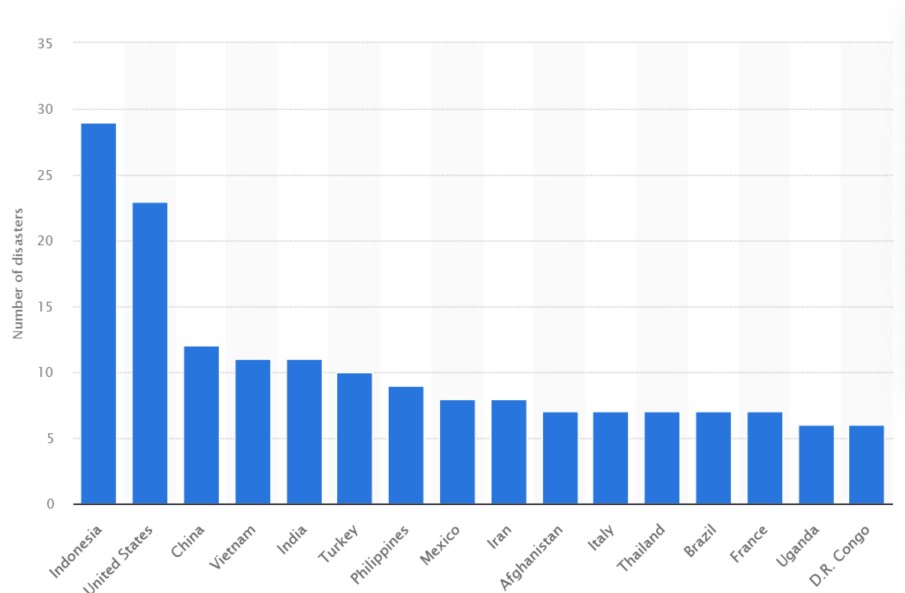

**Figure 2.** Countries with the most natural disasters in 2020 [2].

In addition to natural disasters, there are those caused by mankind (man made), among which are chemical spills, radioactive radiation, accidents by land, air and sea and water pollution, among others. Some examples of these disasters are: In the United States, the Twin Towers attack on 11 September, the Pentagon attack, the nuclear station failure in Japan (Japan's nuclear Fukushima) and the Ukraine's Chernobyl nuclear disaster. Thomas and Kopczak [3] estimate that the frequency of disasters will escalate fivefold in the next 50 years. That coupled with pandemics make it a subject worth studying from all its perspectives, methodologies and approaches.

The other aspect that HSCs raise is the distribution of vaccines and drugs during massive illnesses. Currently with the COVID-19 pandemic, research has turned with greater determination towards this issue. If vaccine distribution is considered, it involves people of all ages, sectors and countries, which makes the distribution of humanitarian

aid and related issues in the humanitarian supply chain an issue with important future projection.

Because this work is a review of the state of the art on metaheuristics applied to HSC, other reviews on the subject were analyzed, from 2014 to date, assessing the approach they used and the results they obtained. Manopiniwes and Irohora [4] review the optimization models used in HSC. These authors found that problems related to relief distribution can be classified into location of the facility, distribution model and inventory, finding that most of the articles focus on the post disaster and very few do so from the stochastic perspective.

Habib et al. [5] study the mathematical contributions by classifying the HSC problem in the location of the facilities, relief distribution and mass evacuation. Contrary to the previous article, they observed that the investigations focus more on the pre-disaster and during the disaster than the post-disaster, but very few articles integrate all the phases due to their complexity, being mostly deterministic and considering unrealistic assumptions. Behl and Dutta [6] reviewed articles published from 2011 to 2017, classifying them in different topics: Research focused on theory, case studies, mathematical models and properties and necessary resources of HSC. They reached the conclusion that it is necessary to add the role of the stakeholders, and the inclusion of qualitative methods or mixed approaches. Chiappetta et al. [7] propose a method for reviewing the literature and systematic selection of HSC management, finding that most of the 87 articles reviewed are theoretical; few of them discuss the location of the disaster, type of disaster or phase and focus mainly on logistics. There are other studies such as Hu et al. [8] that focus on a particular problem such as the optimization of emergency material delivery, concentrating on the optimization model and the applied algorithms presenting both mathematical and heuristic methods.

Hezam and Nayem [9] reviewed articles from 2000 to 2020, and concentrated on mathematical models in HSC, classifying the chain into three main problems: Facility location problem, relief distribution and mass evacuation, in deterministic and non-deterministic problems, also finding that the response phase (disaster) is the most addressed and the post-disaster continues to receive the least attention. Zhang and Cui [10] direct attention to the key decisions in the post-disaster process, focusing on three main topics—facility location, relief material distribution and emergency vehicle routing—concluding that future topics should focus on the storage problem. Furthermore, there is lack of research into emergency transportation and techniques to minimize victims suffering.

After analyzing the previous review articles, to our knowledge there are no published articles that focus on the use of metaheuristics in HSC problems, which is what is addressed in this proposal. MHs are important solution methods in the area because these problems are difficult to solve in reasonable computational time. Knowing the types of problems that have been solved and the most used MHs can help researchers to notice what is missing to do and help to solve real situations.

The article is organized as follows; in Section 2, the definition of HCS and some of its main classifications are shown. Section 3 presents a metaheuristics definition and their classification. In Section 4, the methodology for selecting the analyzed articles is described. Section 5 presents the classification divided into three sections: (1) Type of problem, phases and type of model; (2) time period, objective type and objective function; and (3) metaheuristic classification in HSC. Finally, Section 6 presents the conclusions and the main findings.

## 2. Humanitarian Supply Chain

To understand what an HSC is, different definitions can be considered: Thomas [11] first introduced the concept of humanitarian logistics. He proposed that "humanitarian logistics refers to the process and system of mobilizing human resources, skills and knowledge to help vulnerable groups affected by natural disasters and complex emergencies". Sheu [12] describes emergency logistics as a process of planning, managing and controlling the efficient flows of relief, information and services from the origin to destination to meet

the urgent needs of affected people under emergency conditions. Wassenhove [13] defines a disaster as a disruption that affects a system and its objectives, and considers that the priorities of an HSC are (i) to be a bridge between disaster preparation and response, (ii) in which effectiveness is crucial as well as speed of response in health, food and water distribution, shelter and sanitation; and (iii) it is the most expensive of humanitarian aid efforts. Habib et al. [5] claim that the HCS is "the process of evacuating people from disaster-stricken areas to safe places and planning, implementing and controlling the efficient, cost-effective flow of goods, meanwhile collecting related information from the point of supply to the point of consumption for the purpose of alleviating the sufferings of vulnerable people".

Among the HCS's attributes, it is noted that these kinds of supply chains are clearly unpredictable, turbulent and require flexibility, as well as that they are highly dynamic, innovative and agile [13–15]; sometimes HSCs are slow to respond to the needs of affected people [16]. In principle, locations are often unknown until demand occurs, short lead times drastically affect inventory availability, sourcing and distribution, and it works with very little real-time information. Consequently, making decisions is more complex than in commercial supply chains [15]. In addition to this, HSCs have uncertainty about the scope of the disaster, the number of victims and the number of urgent needs for rescue items [10].

There are different classifications of the supply chain. To begin with, for their attributes in logistics, such as Dasaklis et al. [17] who classify it into four groups: Configuration of the epidemic logistics network, collection of medical supplies, triage operations and other approaches; more generally according to the types of problems that are solved: Facility location and distribution model [4,5,8] and other types of problems are added such as inventory [4] and mass evacuation [5,9].

Kovacs and Spens [15] mention that the central role in logistics responds to whether disasters are natural or man made. In the United States comprehensive emergency management is commonly described in terms of four programmatic phases: Mitigation, preparedness, response and recovery [5,18]. It can also be classified as pre-disaster, post-disaster [5] and the period of the disaster management system. The pre-disaster phase covers the mitigation and preparedness phase; mitigation includes the steps to reduce vulnerability to disaster impact; the response phase addresses immediate threats to minimize economic and human losses, while the recovery phase supports the restoration of all the damage caused by the disaster [5,8]. In Figure 3 the classification of the HSC is shown.

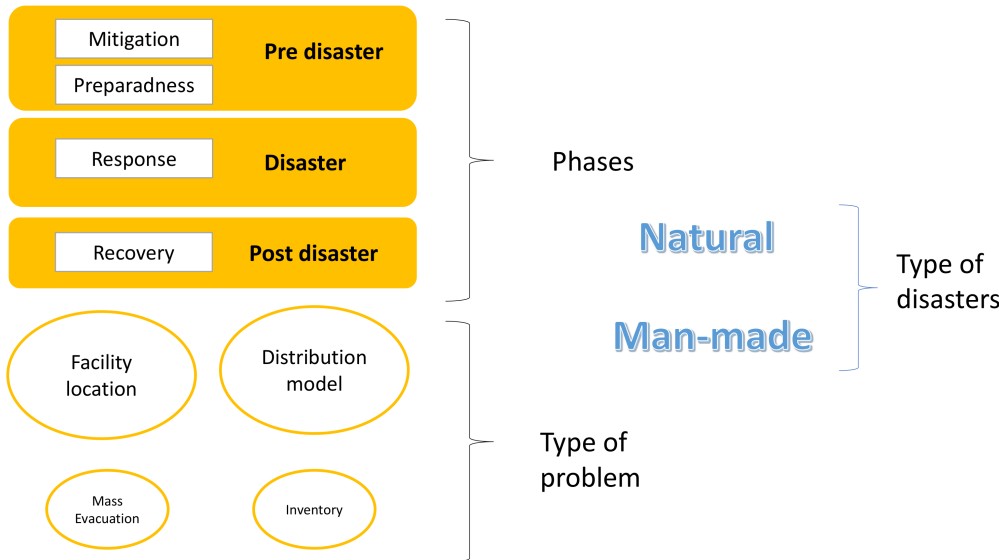

**Figure 3.** HSC Classification.

It is also possible to classify them according to the formulation of the problem in integer and mixed, linear and nonlinear, deterministic and stochastic programming, by the objective function in single or multiple [4,5,9,10]. They can also be categorized according to their solution method in exact, heuristic and metaheuristic methods [8], this classification is shown in Figure 4. Since this focuses mainly on metaheuristics, the following section defines their leading characteristics and methods.

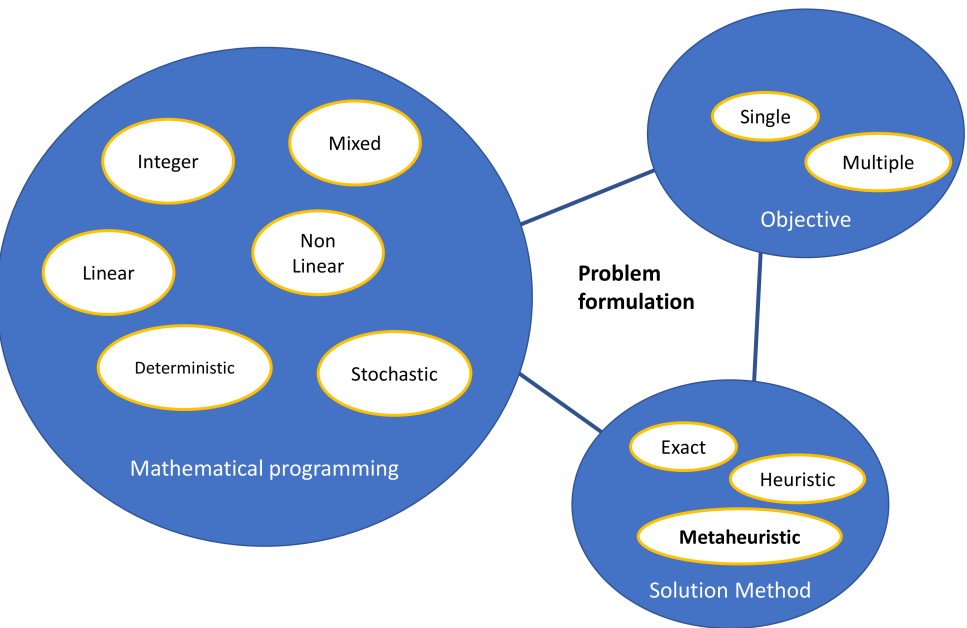

**Figure 4.** Model formulation classification of HC.

### 3. Metaheuristics

An exact algorithm, if it exists, finds an optimal solution together with the test of that optimality or finds that the solution is not feasible or unbounded [19]. The time in which this solution is found must be finite and not very long. On the other hand, heuristics are approximate solution techniques that have been used since the early 1970s, since they began to solve problems known as NP-hard. While various methods have been explored, the most popular is based on local search techniques, which is an iterative process that starts from a feasible solution and in which some modifications (or movements) are made in order to improve the initial solution. The search ends when a local optimum is found for all transformations of the initial solution. The local search depends on the richness of the transformations considered in each iteration. Heuristics apply to a particular problem while metaheuristics (MHs) are more generic.

An MH is a general strategy to guide and control heuristics according to the problems to be solved [20]. They are approximate, they focus on the search space to find a "good enough" solution, parallel implementation is allowed, they start from a local search to advanced search techniques, they incorporate mechanisms to avoid premature convergence and the emerging ones use memory to preserve the search experience [21].

There are different classifications of MH. Osman [22] classifies them into local, construction-based and population-based search; the first works by making small changes to the solution, the second builds solutions by adding a part to an incomplete solution, and the third combines solutions to generate a new one. Gendreau and Potvin [20] classify them as trajectory-based and population-based metaheuristics. Fister et al. [23] divides them into nature inspired and non-natural. The most commonly used metaheuristics for solving problems and some of their main characteristics are described below, presented in alphabetical order.

Ant Colony Optimization (ACO): This is a metaheuristic based on the pheromone trail that follows the behavior of some species of ants. Ants represent stochastic construction

procedures that build candidate solutions using artificial information from pheromones that is based on the ants' search experience and the availability of heuristic information. The cycle consists of three main steps: Problem solutions based on pheromone information and heuristic information are built, solutions are improved through local search, pheromone traces are adapted to reflect the search experience [24].

Genetic algorithm (GA): This is popular in the search for combinatorial solutions that are difficult to solve. It bases its operation on the evolution of the species where each solution represents an individual of a population (chromosome). The recombination of these individuals is what is known as crossing and a small change is made to avoid being trapped in a local optimum or mutation; it is based on the guided search. Subsequently, the objective function is evaluated to know which individual has greater aptitude (fitness) and more opportunity to pass on to the next generation; tournament and elitism are used to bring the best individuals to the next generation [25].

Greedy Randomized Adaptive Search Procedure (GRASP) is a multi-beginning algorithm for combinatorial optimization problems that consists of two phases: Construction and local search. In construction, a feasible solution is generated whose neighborhood is explored until a local optimum is found in the local search and the best solution is saved. In the event that the solution found is not feasible, a repair procedure is applied. There are different techniques to build alternative solutions: Cost disturbances, Lagrangian constructive heuristics, local search in partially constructed solutions, filtering, among others [26].

Iterated Local Search (ILS): This focuses on searching, but not the entire space of candidate solutions, only on solutions that are obtained through an algorithm, typically a local search heuristic. It is an iterative process to find solutions to complex problems, through different disturbances [27].

Large Neighborhood Search (LNS): This algorithm and its adaptive variant (ALNS) have been used to solve transportation and scheduling problems. By using these large neighborhoods it is possible to have access to better candidate solutions in each iteration and find a route more promising. In the LNS an initial solution is gradually improved, destroying and repairing the solution. It is used for combinatorial problems where a neighborhood or set of possible solutions is found, using an improved algorithm that selects the best solution in the neighborhood. Adaptive allows multiple destruction and repair methods in the same search process [28].

Scatter Search (SS): This is an evolutionary metaheuristic that explores solution spaces through a set of reference points operating on a small set of solutions using limited randomness. The benchmarks refer to good solutions to the problem and are not necessarily restricted to the objective function. Three steps are used: To combine, to improve and to update the solutions. These same principles are used by GRASP and Path Rethinking [26].

Simulated Annealing (SA): This is based on the analogy with the annealing of materials; through the Monte Carlo approach it is achieved that the method is not trapped in a local optimum. It is one of the simplest and best known to deal with global optimization problems [29].

Swarm Intelligence (SI): This is a discipline that studies the collective behaviors of natural systems, insects and animals. This concept is inspired by natural phenomena that have the ability to solve problems that are a challenge for computational techniques. The system or population is complex and highly organized, which results in interactions between the individuals of the colony, as well as with the environment, following simple rules. Biological examples are emulated, such as ants looking for their route, fish that swim to escape predators, bees that tell their peers where food is. These include the Particle Swarm Optimization, which is a metaheuristic inspired by the behavior of animals, insects and humans; each individual represents a potential solution that seeks to improve their position by taking information collected by themselves and by their neighbors. Through a random disturbance, the position can be adjusted according to the speed of the particle [30].

Tabu Search (TS): This is used to solve combinatorial problems, the basic principle is to pursue the local search and if no improvement is found, avoid returning to those solutions that have been saved in a tabulist. Glover identifies it as a metaheuristic and the search space depends on the problem to be solved [31].

Variable Neighborhood Search (VNS): This is a metaheuristic to solve combinatorial problems whose idea consists of systematic neighborhood change in a descending phase, in order to find an optimum and a disturbance phase to leave the corresponding valley [19].

Some recently proposed metaheuristics to address engineering problems are described below.

Barnacles Mating Optimizer (BMO): This is based on the mating behavior of barnacles. BMO is classified in the group of evolutionary algorithms. A barnacle's mating group consists of all the neighbors within reach of its penis and all its potential competitors per mate. One of the main characteristics of barnacles is their long penises, which is considered in the selection process, similar to GA. The reproduction process is based on the characteristics of inheritance or genotypic frequency of the parents of barnacles when producing the offspring. The BMO has been applied to optimal reactive power dispatch problems achieving competitive results compared to other metaheuristics [32].

Competitive Swarm Optimizer (SCO): This is based on Particle Swarm Optimization (PSO) and uses a competition mechanism in pairs, where the particle that loses the competition will update its position by learning from the winner. CSO has shown to perform surprisingly well on large-scale optimization problems [33].

Falcon Optimization Algorithm (FAO): This algorithm is inspired by the flight rules that falcons follow to hunt their prey. Furthermore, the reference functions for FAO have characteristics such as continuous, separable, differentiable, scalable, unimodal or multimodal. The unimodal functions allow the exploitation of the optimization techniques and the multimodal ones allow to evaluate the exploration and the avoidance of local optimum. This algorithm has shown competitive processing time and achieves good single-objective results when applied to shell-and-tube and plate-fin heat exchanger problems [34].

Hybrid Harris Hawks–Sine Cosine Algorithm (hHHO-SCA): This is a hybrid algorithm based on the Harris Hawks optimizer and the sine–cosine algorithm, which aims to accelerate the global search phase. This method has been successfully tested for highly constrained, nonlinear and non-convex engineering design problems [35].

Manta Ray Foraging Optimization (MRFO): This is based on intelligent behaviors of manta rays. This algorithm has three foraging operators, including chain foraging, cyclone foraging and somersault foraging. MRFO mimics three stingray feeding strategies that are mathematically modeled as a new alternative optimization approach to address real-world engineering problems. Experimental results show that this algorithm is reliable and effective in solving complex problems [36].

Owl Optimization Algorithm (OOA): This is a swarm intelligence-based metaheuristic. It is inspired by owls' decoy behavior. Owls use different strategies to avoid predator attacks. The OOA mimics the Owl's movement for the search process. OOA has been applied to solve heat exchanger problems, achieving better results than other algorithms proposed in the literature [37].

Pathfinder Algorithm (PFA): This is a swarm-based method and is proposed for continuous optimization with different structure. The PFA method mimics the collective movement of swarms with the use of hierarchy between the leader and other members of the swarm. The MOPFA algorithm is designed to solve multi-objective engineering problems [38].

Poor and Rich Optimization (PRO): This algorithm is inspired by a real social phenomenon, related to human behavior in obtaining wealth and improving their economic status. The algorithm imitates the behavior of two categories of individuals, those of the rich class and those of the poor class; where the members of the poor group try to improve their status and reduce the gap by learning from the rich, while the individuals of the rich category try to increase the class gap by observing the poor and acquiring wealth. The pop-

ulations were randomly generated. According to the experimental results, this algorithm has shown better performance in engineering problems by finding optimal values of the parameters compared to the results of algorithms proposed in the literature [39].

Search and rescue optimization algorithm (SAR): This is a method of solving engineering-constrained optimization problems. This algorithm mimics scanning behavior in search and rescue operations. In addition, to handle constraints it uses the $\varepsilon$-constrained method [40].

Supply–Demand-based Optimization (SDO): This is a metaheuristic swarm-based method, inspired by supply–demand mechanism, where the principal characteristic mimics both the demand relation of consumers and the supply relation of producers. SDO has been proposed to address constrained engineering problems, achieving better results than the algorithms proposed in the literature [41].

There are other recent MH proposals such as Water Waves Optimization; Clonal Selection Algorithm; Gases Brownian Motion Optimization; Music Based Metaheuristics, Harmony Search and Method of Musical Composition; Physic Based, Gravitational Search Algorithm; Social and Sport based; Teaching Learning Based Optimization, League Championship Algorithm; among others [21]. Additionally, there are highly complex scenarios where pure metaheuristics do not give as good results as combining different techniques; they hybridize, taking the best characteristics of each one, which is why they give better results. According to Raidl et al. [42] the following can be hybridized, (a) metaheuristics with metaheuristics, (b) metaheuristics with problem-specific algorithm simulations, (c) metaheuristics with OR techniques, exact as dynamic programming, linear and non-linear programming, with other heuristics such as neural networks, fuzzy logic, statistical techniques and (d) metaheuristics with human interactions.

Moreover, there are MHs generated to solve multi-objective problems, especially those objectives that are in conflict [21]. Non-dominated Sorting Genetic Algorithm (NSGA) and NSGA-II incorporate standard genetic algorithms and non-dominated sorting to solve problems with multiple objectives under different constraints. NSGA-II includes elitism, uses diversity as a preservation mechanism and its complexity is low compared to NSGA. Multi-Objective Particle Swarm Optimization with Crowding Distance (MOPSO) is an MH with two different approaches; the first consists of considering each objective function separately managed by a particle; in each one the best positions are selected, the difficult part being guiding the particles to Pareto-optimal solutions. The second approach is the evaluation of each particle by all objective functions based on the Pareto-optimality concept, which produces non-dominated solutions to guide the particles. In Figure 5, the classification of metaheuristics is shown.

The following section explains the methodology used to classify the articles.

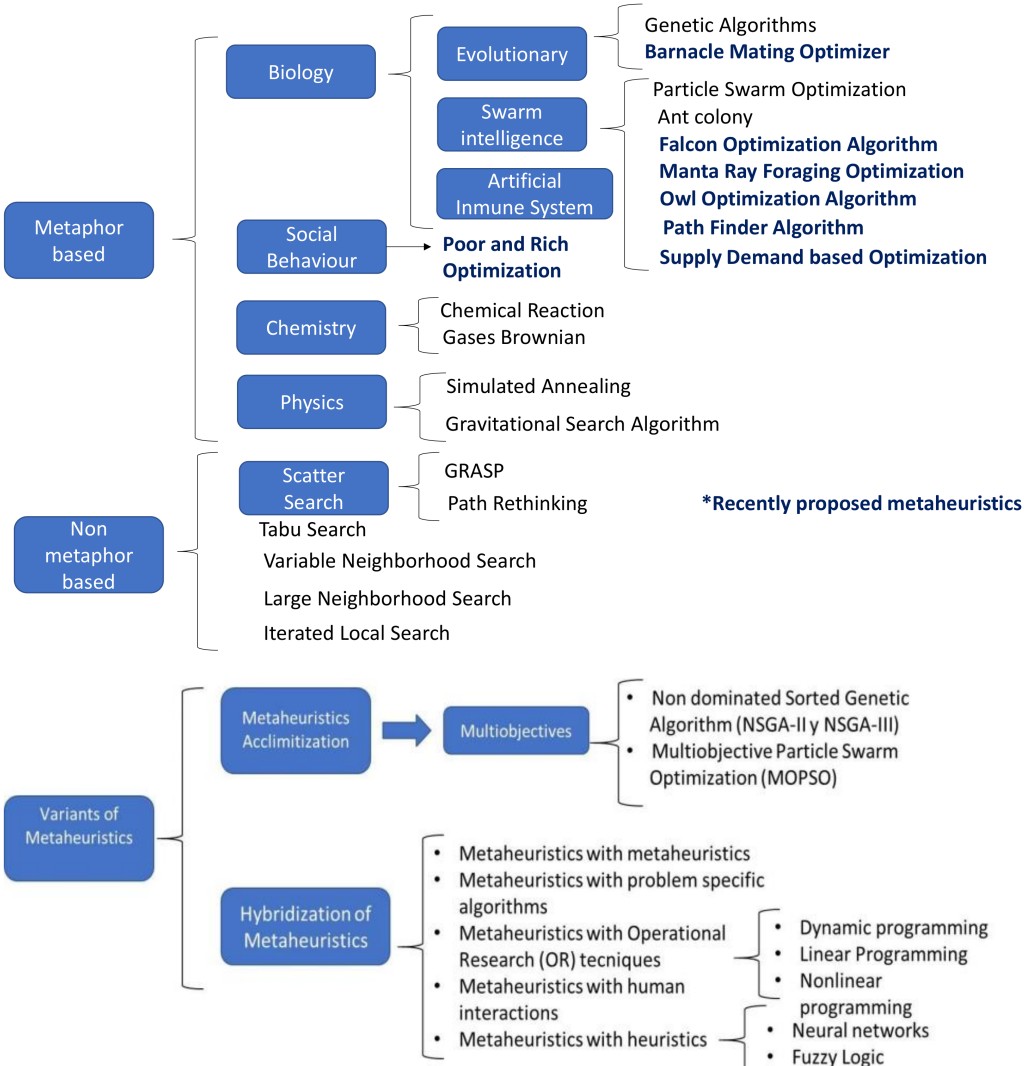

**Figure 5.** Metaheuristic classification, based on Abdel-Basset et al. [21].

## 4. Methodology

To achieve the review of the state of the art, the methodology of the systematic literature review was followed. The purpose is to summarize the research carried out on a particular topic and to find important elements that help to generate knowledge. Tranfield et al. [43] consider four aspects, which are detailed below:

(a)   Planning: At this stage, the research questions were asked to avoid ambiguous answers. The questions generated were the following: RQ1. How are the HSC problems that have been solved from Metaheuristics since 2016 classified? RQ2. What is the gap found to accomplish future research in Metaheuristics in HSC?

(b)   Searching: At this stage, articles were browsed in three databases—Web of Science, Scopus and Google Scholar—using the following keywords, "optimization", "humanitarian supply chain", "relief supply chain", from 2016 to date. It is necessary to mention that "metaheuristics" was not used because it considerably decreased the number of articles found. There were 120 found in Web of Science, 289 in Scopus and 1680 in Google Scholar.

(c)   Screening: In this phase, the inclusion and exclusion criteria were established. Inclusion: Articles that used as a metaheuristic for the HSC solution were selected, all were peer-reviewed research articles from 2016 to date. Exclusion: Articles that do not use a metaheuristic for the solution and refer to the administration of HSC are excluded from this research. Duplicate articles, those that are conference articles and review

articles were not considered for classification. After reading the articles, 80 articles were selected.

(d)  Extraction: In this phase, the selected articles were read and analyzed to classify them according to the characteristics of the HSC that are detailed in the next section.

## 5. Classification

This section presents the articles analyzed in order to answer the research questions.

### 5.1. Type of Problem

Facility location: This refers to determining where shelters, medical centers, warehouses and/or distribution centers for emergency materials should be located [10]. It also deals with the problem of using existing ones. This problem is vital because it can mitigate human pain [9] when meeting the demand at the minimum cost at the maximum level of service [5]. Among the articles found in the review, Shavarani [44] finds the best location of the refuge and drone centers using the multilevel Facility Layout Problem, using the closest neighborhood method. In 2021 Shavarani et al. [45] had already solved the problem using NSGA-II. Madani [46] proposes a network with multiple links that considers the location of hospitals, warehouses and hybrid centers, using the NSGA-II with Simulated Annealing and with Variable Neighborhood Search. In Khorsi et al. [47], a location and routing problem is presented to deliver goods to victims considering uncertainty in demand; it is used with a multi-objective grouping algorithm to find Pareto-optimal solutions.

Distribution: This problem refers to the design of the network and its location, the type of transport and its capacity, deciding whether the fleet is homogeneous or not, last-mile operations and what refers to the transportation of elements in HSC [9]. Distribution can also include transporting injured people to hospitals. Razavi [48] presents a distribution model for blood transfusion using a multi-objective mathematical model and solving through a genetic algorithm. Davoodi and Goli [49] minimize the arrival time of aid vehicles to disaster areas using hybrid bender decomposition and variable neighborhood search. Boonmee et al. [50] present a post-disaster waste management model, including a recycling decision model. A mixed integer programming model is presented that is solved with Particle Swarm Optimization and Differential Evolution. Ghaffarri et al. [51] present a network for the distribution of medical items that includes local providers and regional distribution centers and points of demand; the situation is modeled using mixed-integer programming that is solved with Particle Swarm Optimization. In the problem of Beiki et al. [52] a logistic problem of multiple vehicles is shown considering levels of satisfaction and environmental conditions; for optimization a Genetic Algorithm is used. Macias et al. [53] solve a problem of Unmanned Aerial Vehicles to be used in the HSC, a novel multi-stage model is designed and a routing algorithm that is solved through Large Neighborhood Search. Other distribution works are Mamashli et al. [54], Talebi and Salari [55] and Molladavoodi et al. [56].

Inventory: Some authors include in the facility location problem the integration of inventories in the aid distribution from the distribution centers to the warehouses considering the shortages and the penalty in the total cost. Rezaei et al. [57] develop a bio-objective optimization model to operate a supply chain of car fuel in earthquake areas. The objective function includes the unmet demand penalty and inventory cost that is solved with the Grasshopper Optimization Algorithm. Hajipour et al. [58] present a distribution plan and inventory during a disaster scenario through a nonlinear bio-objective mathematical model that is solved through Multi-Objective Vibration Damping Optimization (MOVDO) and NSGA-II.

Mass evacuation: This problem deals different decisions: Where the evacuation points are concentrated, selection of evacuation transport, route capacity, traffic management, considering whether the transport will be public or private, in addition to deciding whether to evacuate without evacuation notice or to have a period of between 24 to 72 h [5,9]. Molina et al. [59] consider the Multi-Objective Trained Vehicle Routing Problem for the evacuation

of people affected in the disaster; the Multiple Start Algorithm with Smart Neighborhood Selection is used. Mollah et al. [60] work on the evacuation of people during a flood; two algorithms are proposed, one of integer mixed programming and a genetic algorithm. Jha et al. [61] presents a disaster evacuation chain with two echelons—evacuation fields and affected—modeling the problem through mixed integer programming that is solved through NSGA-II. In [45] the transfer of affected people is considered using forward and backward multi-objective and is resolved with NSGA-II.

### 5.2. Model Type and Phases

Deterministic: In deterministic problems, the input parameters are known and constant over time [9]. In the articles found for the classification 41 of 78 are deterministic. As an example, the following works are analyzed. Decerle et al. [62] present the routing and programming of caregivers to the patient's home using the ant colony algorithm. Frifrita et al. [63] also consider a problem of assigning caregivers to patients using synchronization and time window restrictions, the General Variable Neighborhood Search is used to solve. Sujaree and Sammattapong [64] use the hybrid artificial chemical reaction algorithm to find the routes and deliver vaccines. Noham and Tzur [65] design a resource distribution network in the event of a disaster; a mathematical model is incorporated and solved through Tabu Search.

Non-Deterministic: In this type of approach, the parameters are uncertain and can be modeled through a probability distribution (stochastic models) [9] or through discrete scenarios in intervals (fuzzy logic). In the classification of articles, 37 out of 70 are presented in this modality. In the work of Huang and Song [66] the emergency logistics and the routing problem are considered, where the demands of the affected areas and the travel times are expert estimations, for which variables with uncertainty are considered, and are solved with a cellular genetic algorithm. Babaei and Shahanaghi [67] plan a flow of relief items under uncertain conditions such as demand using fuzzy logic and solving through NSGA-II. Bozorgi-Amiri et al. [68] present a plan to provide critical items to affected people considering a stochastic programming model that integrates pre- and post-disaster decisions and is resolved through Multi-Objective Particle Swarm Optimization (MOPSO).

Pre-disaster: As already mentioned, this stage includes the mitigation and preparation stages; in this phase the necessary measures are taken to reduce the severity of the problem, a strategic approach is used to choose relief or distribution centers, people are evacuated to a safe place [5]. Of the articles found for the classification, only 14.10% did so in the pre-disaster stage. Adarang et al. [69] consider a problem of location and routing to provide medical services in order to plan and manage transportation under uncertainty, using the shuffled frog jump algorithm and the NSGA-II. In the work of Akdogan et al. [70], the location of emergency vehicles is studied through a queueing model, a mathematical model is used to minimize response time and a Genetic Algorithm is also used. Shavarani [44] used the NSGA-II to address a facility location problem to minimize the total relief items supply chain cost; the model type is non-deterministic. Mardaninejad and Nastaram [71] studied a facility location problem through a deterministic model to minimize the distance and fixed cost of equipping a temporary accommodation center; they address a study case and used SA, PSO, ICA, ACO, ABC, FA and LAFA algorithms. Nayeri et al. [72] considers a mass evacuation problem through a multi-objective deterministic model to minimize the completion time of relief operation, using GA and PSO algorithms. On the other hand, Hasani and Mokhtari [73] proposed a multi-objective non-deterministic model to study a facility location and inventory problems through NSGA-II and PSO algorithms to minimize total coverage by the relief network, minimize total cost and minimize the maximum risk of total demand nodes.

Post-Disaster: This includes the recovery phase. Once the damage is calculated, the related transportation is repaired and relief items are generally distributed from a distribution center to the affected areas; it also considers calculations such as the number of people affected, how many distribution centers or shelters must be installed and how many

items must be delivered. In Wu et al. [74] goods are distributed in an emergency supply chain, the problem is known as routing of trained vehicles considering fair distribution; to solve the problem, a hybrid algorithm of Ant Colony and Variable Neighborhood Search Algorithm is proposed. Vahdani et al. [75] propose a multi-objective nonlinear integer model to locate distribution centers and carry out aid to damaged areas through vehicle routing, using NSGA-II and MOPSO. Goodarzian et al. [76] propose a model of production, distribution, inventory and location in a sustainable supply chain; three metaheuristics are tested: Ant colony, fish swarm and firefly algorithm. Wang et al. [77] propose a model of assistance and food delivery teams through demand points, using an integer programming algorithm and two hybrid metaheuristics. Caballero Morales et al. [78] propose an MH based on K-means clustering and a micro genetic algorithm to estimate a search interval for help centers; they also use GRASP to establish 260 help centers in 3837 at-risk communities in Veracruz, Mexico. Mardaninejad and Nastaram [71], by transferring earthquake victims to safe places and medical centers by selecting candidate locations, formulate the optimal accommodation to perform using GAMS and seven metaheuristics such as Simulated Annealing and Ant Colony, among others.

There are other approaches that handle the pre-disaster and post-disaster phases in an integrated manner. For instance, Tavaana et al. [79] propose a network that considers the location of warehouses managing the inventory of perishable products in the pre-disaster phase and the routing of vehicles in the post-disaster phase; NSGA-II with and without reference point are used to solve this mixed-integer programming problem. Mohammadi et al. [80] develop a response plan to provide items to affected people, a stochastic multi-objective model is proposed that integrates pre and post-disaster decisions, and MOPSO is used to solve it. Shi et al. [81] study the vehicle routing scheduling problem for home health care companies; uncertainty in demand is considered as a fuzzy variable and a hybrid genetic algorithm with stochastic simulation is proposed. Su et al. [82] with the problem of how to place emergency resource agencies in multiple ways to attend to incidents and reduce economic losses, present a restricted programming model and a differential evolution heuristic based on a search algorithm is developed. Sharma et al. [83] propose the integration of different techniques to locate blood banks in conditions during and post-disaster; the distance between hospitals is minimized and Tabu Search is used to calculate the optimal number of centers.

In Figure 6 the classification and the number of articles found in each of the areas are shown. This figure illustrates more jobs in the post-disaster area and integrated post and pre-disaster. The most commonly addressed problems are deterministic due to the complexity of non-deterministic models; however, the advantage of the latter is to consider the inherent randomness complexity of the real cases. The jobs that appear the most are those related to solving distribution problems, followed by the facility location problem (or a combination of both); there is still a smaller number that is related to inventories and mass evacuation; with this Research Question 1 (RQ1) is resolved.

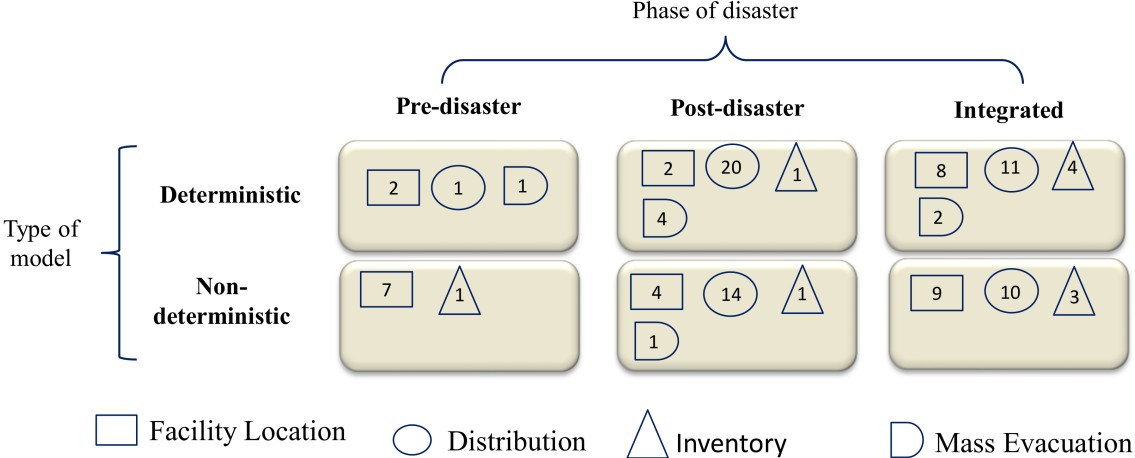

**Figure 6.** Classification of problem type, model type and phase of a disaster.

### 5.3. Time Period, Objective Type and Objective Function

Objective function: Concerning the objective function it can consider costs, time or distance [8]; it depends on the problem to be solved; both the time and the distance are important to give a quick response to the victims. Another goal is to satisfy as many people as possible and to ensure fairness. The objectives of maximizing and minimizing that were found in the reviewed articles are listed below.

Minimize:

- Cost or distance, refers to using vehicles to bring aid to disaster areas [55,59,75,84,85].
- Costs of help center locations [55,78].
- Customer waiting times taking fairness into account [74,86].
- Cost of transporting the population out of danger zones [60].
- Penalty costs for unsatisfied demand [57,87–90].
- The infection possibility [91].
- The level of discontent of facing injustice [48].
- The financial effects and variable costs [50].
- The number of injured people who have not been attended to [52].
- Environmental aspects when relief items are carried [54].
- Completion times [72,92].

Maximize:

- Total care coverage [48,80,93].
- Average response for time-care [91,93].
- The profit [50].
- System reliability [46].
- The death toll [94].
- The number of undamaged items received in warehouses [58].

Additionally, from the reviewed papers, 44 are bi- or multi-objective and 36 have a single objective (RQ1). Regarding bi-objective and multi-objective, Rezaei et al. [57] proposed a bi-objective deterministic model to address the inventory problem in the post-disaster phase for designing a fuel supply chain network. They used MOEA, NSGA-II and MOPSO algorithms to minimize the penalties due to both delayed and unsatisfied fuel demands and the difference between the satisfied demand in different earthquake-affected areas. Abazari et al. [84] addressed facility location, distribution and inventory problems in the integrated phase of disaster for distributing relief items with uncertain parameters. They proposed a multi-objective non-deterministic model to minimize distance travelled by relief items, the travelling time from facility to demand location and the total quantity of perished items. To solve the proposed model, they used a Grasshopper Optimization Algorithm (GOA). Jiang et al. [91] formulated an optimization model on fresh agri-product

emergency supply to study the response time, infection risk and transportation resources. They proposed a multi-objective deterministic model to study a distribution problem in the post-disaster phase to minimize average response time, the infectious possibility and transportation resource utilization. An improved genetic algorithm based on solution features (IGA-SF) was used to solve the model. On the other hand, Sadeghi et al. [95] solved a distribution problem through a single-objective deterministic model in the integrated phase of a disaster to maximize the demand coverage and reduce the rescue time, using the simulated annealing (SA) algorithm to solve the model. Ransikarbum and Mason [96] formulated a single-objective deterministic model in the post-disaster phase to address a distribution problem to minimize the total cost of distribution using the NSGA-II algorithm.

Regarding the period, 19 are multi-period and 61 are single period; this may derive from the complexity of solving a multiple period problem (RQ1). Some of the articles reviewed are as follows. Babaei and Shahanaghi [67] presented a multi-objective non-deterministic model with a single phase; facility location and mass evacuation problems were studied; they analyzed a multi-level location–allocation routing emergency problem in uncertain conditions to cover all demands existing in the network. Vahdani et al. [75] propose a multi-objective, multi-period, multi-commodity model; facility location and distribution problems were addressed in the post-disaster phase to minimize travel time and total cost and increase the reliability of routes; to solve the problem, NSGA-II and MOPSO algorithms were used. In the paper of Qi and Hu [97] emergency cold chain logistics scheduling is considered, including the loss of the vehicle, refrigeration consumption and damage of goods over time; a multi-period deterministic model to minimize the total cost of distribution was proposed; to solve the problem, ant colony system (ACS) and Pareto local search (PLS) algorithms were implemented; they found that ACS and PLS algorithms have strong applicability. On the other hand, the paper of Wu et al. [74] proposes an ant colony optimization-variable neighborhood search (ACO-VNS) algorithm to study a distribution problem through a single-phase non-deterministic model to minimize waiting time of customers taking peer-induced fairness into account; they found that hybrid algorithm achieves better performance in a short time. Zahedi et al. [98] address a distribution problem through a single-phase, multi-objective, non-deterministic model to minimize the starting time of the suspected case with the lowest priority by each ambulance and penalty time of visitation and the critical response time. Simulated annealing (SA), Social Engineering Optimization (SEO) and Particle Swarm Optimization (PSO) algorithms were used to solve the problem; they addressed a real case and implemented their proposal, achieving a decrease in infections.

### 5.4. Metaheuristics Classification in Humanitarian Supply Chains

Regarding metaphor-based Metaheuristics classification, 56 publications were found for HSC problems. Of which, 44 belong to metaheuristics from the area of biology, 12 to the area of physics and one to the area of social and sport. The most used were Genetic Algorithm (GA), Particle Swarm Optimization (PSO) and Simulated Annealing (SA) with 9, 12 and 12 applications, respectively. Regarding the GA metaheuristic, the problems studied were facility location (3), distribution (3) and mass evacuation (2). The problems addressed with the PSO algorithm were facility location (3), distribution (9), inventory (1) and mass evacuation (1). Likewise, the problems addressed with the SA algorithm were facility location (7), distribution (8) and mass evacuation (2), see Figure 7.

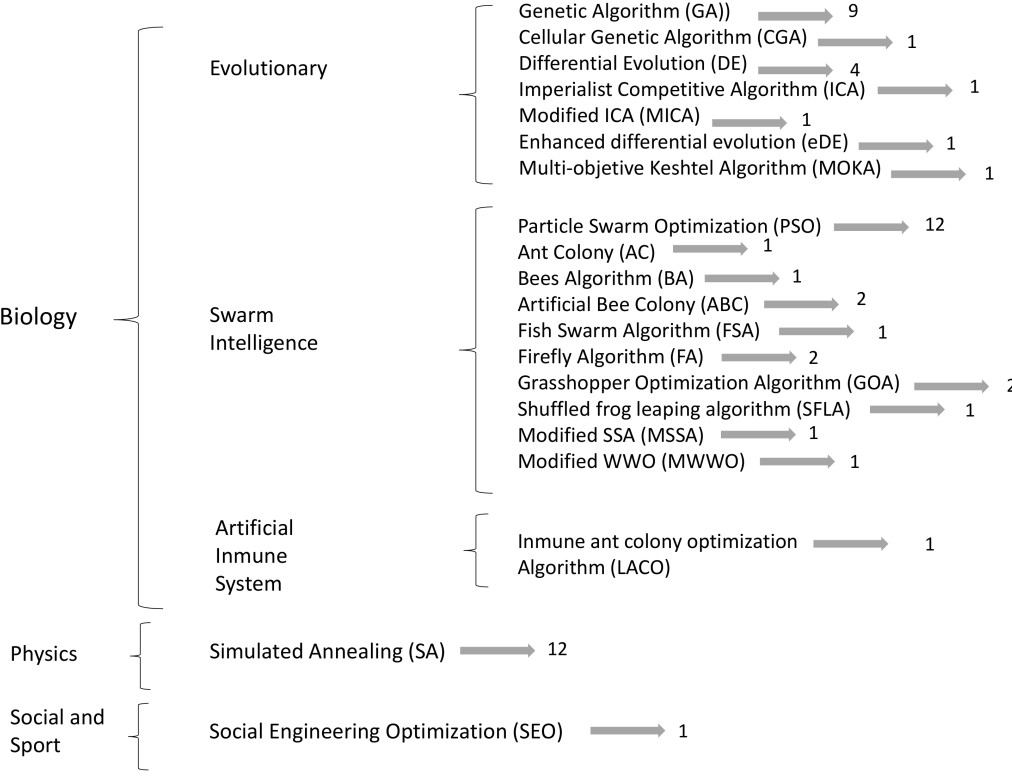

**Figure 7.** Metaphor-based metaheuristics.

On the other hand, with reference to the Non-Metaphor-based Metaheuristics classification, 22 articles were found; the most used metaheuristics were Search Algorithm (SA) with 12 applications and Greedy Randomized Adaptive Search (GRASP) with three applications. The problems addressed with SA were facility location (7), distribution (8), and mass evacuation (1). With GRASP, four distribution problems were addressed, see Figure 8.

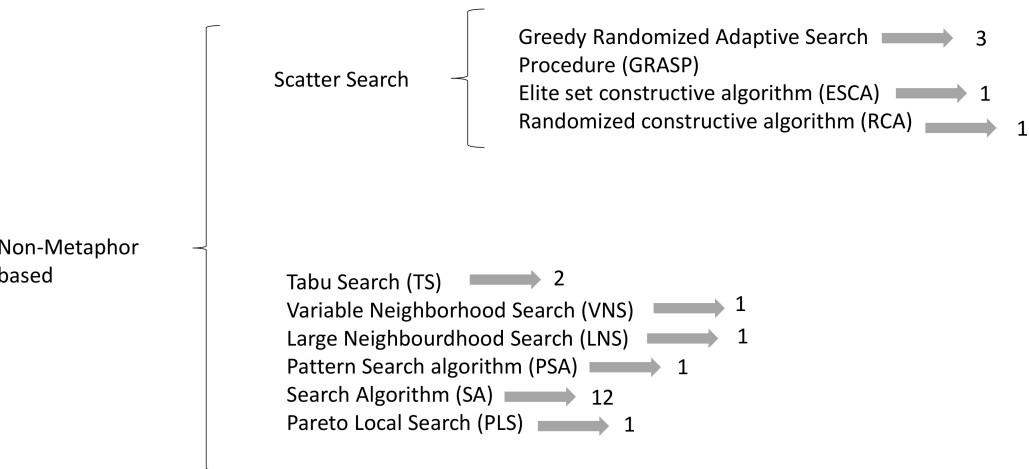

**Figure 8.** Non metaphor-based Metaheuristics.

Regarding the multi-objective metaheuristics, 30 works were reviewed; the most used were Non-dominated Sorting Genetic Algorithm-II (NSGA-II) with 15 applications and Multi-Objective Particle Swarm Optimization (MOPSO) with five. The problems addressed with NSGA-II were facility location (9), distribution (6), inventory (3) and mass evacuation (1). Furthermore, the problems addressed with MOPSO were facility location (3), distribution (4) and inventory (1). Finally, in the matter of hybridization of

metaheuristics, 12 applications were found; however, no predominant one was found; the problems addressed were distribution (9) and mass evacuation (1); for further reference see Figure 9.

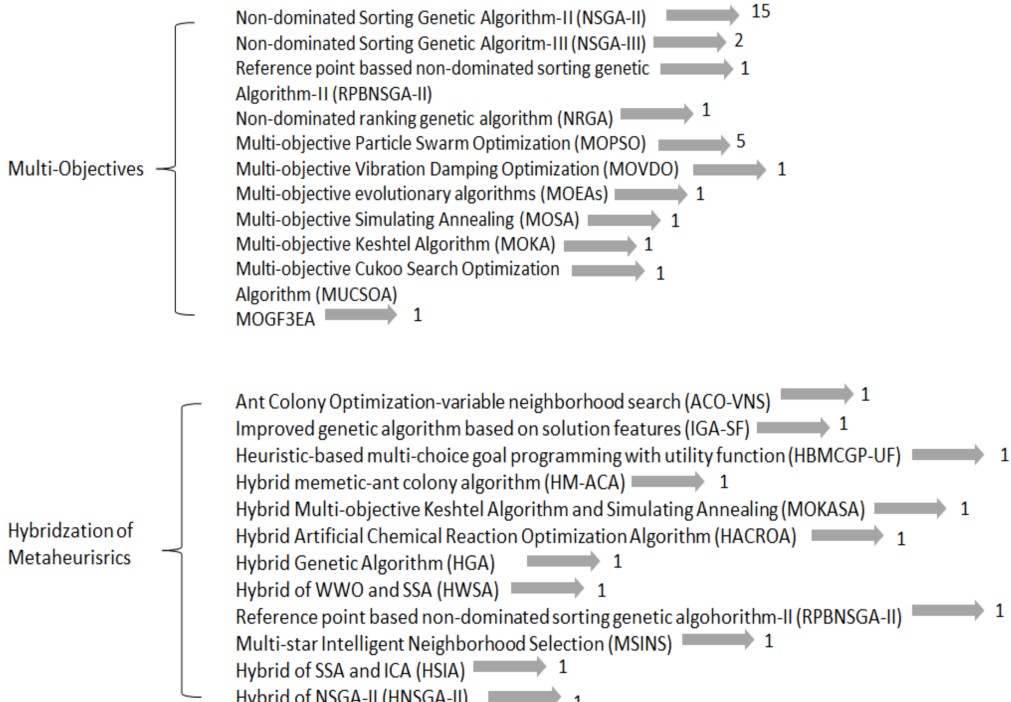

**Figure 9.** Multi-objective and Hybridization of Metaheuristics.

In Table 1 all the analyzed papers are classified.

Table 1. Paper analysis, * means that the article meets this characteristic.

| Authors | Phase of Disaster | | | Model Type | | Problem Type | | | | Metaheuristic Algorithm | Time Period | | Objective Type | | Objective Function Minimize (−) Maximize (+) (the Number Indicates the Consecutive of the Objectives) | Results | |
|---|---|---|---|---|---|---|---|---|---|---|---|---|---|---|---|---|---|
| | Pre-Disaster | Post-Disaster | Integrated | Deter-ministic | Non-Deterministic | Facility Location | Distri-bution | Inventory | Mass Evacuation | | Single | Multi | Single | Bi/Multi | | Random Data | Case Study |
| Tavanaa et al. [79] | | | * | * | | * | * | * | | NSGA-II, RPBNSGA-II | * | | | * | (1) (−) Total cost of procurement and preparation pre-disaster phases, (2) (−) the total relief operational cost on post-disaster, (3) (−) the total operational relief time on post-disaster. | | * |
| Molina et al. [59] | | * | | * | | | | | * | MSINS | | * | | * | (1) (−) The number of vehicles, (2) (−) total traveling cost, (3) (−) the maximum latency. | | * |
| Babaei and Shahanaghi [99] | | * | | | * | * | | | * | SA | * | | | * | (1) (−) Total cost of establishing the emergency location, (2) (−) the cost of constructing the path, (3) the number of required ambulances in each scenario. | * | |
| Wu et al. [74] | | * | | | * | | * | | | ACO-VNS | * | | * | | (−) The sum of customers waiting times taking peer induced fairness into account. | * | |
| Vahdani et al. [75] | | * | | * | | * | * | | | NSGA-II MOPSO | | * | | * | (−) The travel time and total cost and increases reliability of the routes | * | |
| Mollah et al. [60] | | * | | * | | | * | | * | GA | | * | * | | (−) Total cost for transporting population and relief-kits and penalty cost associated with one un-evacuated in-need population. | | * |

Table 1. *Cont.*

| Authors | Phase of Disaster | | | Model Type | | Problem Type | | | | Metaheuristic Algorithm | Time Period | | Objective Type | | Objective Function Minimize (−) Maximize (+) (the Number Indicates the Consecutive of the Objectives) | Results | |
|---|---|---|---|---|---|---|---|---|---|---|---|---|---|---|---|---|---|
| | Pre-Disaster | Post-Disaster | Integrated | Deter-mini-stic | Non-Determi-nistic | Facility Location | Distri-bution | Inventory | Mass Evacua-tion | | Single | Multi | Single | Bi/Multi | | Random Data | Case Study |
| Rezaei et al. [57] | | * | | * | | | | * | | MOEAs, NSGA-II, MOPSO | * | | | * | (1) (−) The penalties due to both delayed and unsatisfied fuel demands, (2) (−) the difference between the satisfied demand in different earthquake-affected areas. | | * |
| Abazari et al. [84] | | | * | | * | * | * | * | | GOA | * | | | * | (1) (−) Distance traveled by relief items, (2) (−) RC establishing cost, (3) (−) the maximum traveling time from facility to demand location, (4) (−) the total quantity of perished items. | | * |
| Mohammadi et al. [80] | | | * | | * | * | * | | | MOPSO, PSO | * | | | * | (1) (−) Total expected demand coverage, (2) (−) the total expected cost, (3) (−) the difference in the satisfaction rates between nodes. | | * |
| Jiang, Bian and Liu [91] | | * | | * | | | * | | | IGA-SF | * | | | * | (1) Average response time, (2) the infectious possibility, (3) the transportation resource utilization | | * |
| Goodarzian et al. [76]. | | * | | | * | * | * | * | | ACO, FSA, FA | | * | | * | (1) (+) Social factors, (2) (−) the cost of establishing DCs, inventory holding, transportation cost, production cost, (3) (−) the maximum unmet demand. | | * |

Table 1. *Cont.*

| Authors | Phase of Disaster | | | Model Type | | Problem Type | | | | Metaheuristic Algorithm | Time Period | | Objective Type | | Objective Function Minimize (−) Maximize (+) (the Number Indicates the Consecutive of the Objectives) | Results | |
|---|---|---|---|---|---|---|---|---|---|---|---|---|---|---|---|---|---|
| | Pre-Disaster | Post-Disaster | Integrated | Deterministic | Non-Deterministic | Facility Location | Distribution | Inventory | Mass Evacuation | | Single | Multi | Single | Bi/Multi | | Random Data | Case Study |
| Wang et al. [77] | | * | | | * | | * | | | ABC | * | | * | | (−) Total service completion time among all demand points. | | * |
| Jha, Acharya, and Tiwari [61] | | * | | * | | | * | | * | NSGA-III | * | | | * | (1) (−) The cost of set-up, procurement, transportation between supplier and relief camps, (2) (−) gap between demand and supply of the relief chain. | * | |
| Noham and Tzur [65] | | | * | | * | * | | * | | TS | * | | * | | (+) The ratio of units distributed to their delivery time. | * | * |
| Razavi et al. [48] | | * | | | * | | * | | | GA | * | | | * | (1) (−) The cost of the blood supply chain, (2) (−) the maximum degree of discontent with unfairness among affected areas, (3) (+) coverage of the demand of blood in field hospitals. | * | |
| Davoodi and Goli, [49] | | | * | * | | * | * | | | VSN | * | | * | | (−) The maximum interval times of vehicles to depot R + 1. | | * |
| Shavarani [44] | | * | | | * | * | | | | NSGA-II | * | | * | | (−) The total travel distance to meet the demand on each point. | | * |
| Boonme et al. [50] | | * | | | * | | * | | | PSO, DE | * | | | * | (1) (−) The financial effects of the fixed and variable costs, (2) (+) revenue from sellable waste. | * | |

Table 1. *Cont.*

| Authors | Phase of Disaster | | | Model Type | | Problem Type | | | | Metaheuristic Algorithm | Time Period | | Objective Type | | Objective Function Minimize (−) Maximize (+) (the Number Indicates the Consecutive of the Objectives) | Results | |
|---|---|---|---|---|---|---|---|---|---|---|---|---|---|---|---|---|---|
| | Pre-Disaster | Post-Disaster | Integrated | Deter-ministic | Non-Determi-nistic | Facility Location | Distri-bution | Inventory | Mass Evacua-tion | | Single | Multi | Single | Bi/Multi | | Random Data | Case Study |
| Ghaffari et al. [51] | | * | | * | | | * | | | PSO | | * | * | | (−) Total weighted completion times of services at hospitals. | * | |
| Eskandari-Khanghahi et al. [100] | | | * | | * | * | * | | | SA | | * | | * | (1) (−) The total environmental impacts, (2) (+) The social impacts, (3) (−) the total variable and fixed cost in the network. | * | |
| Beiki et al. [52] | | * | | | * | | * | | | NSGA-II, MOPSO | * | | | * | (1) (−) The maximum number of the unserved injured people, (2) (−) the sum of cost. | | * |
| Macias et al. [53] | | * | | * | | | * | | | LNS | | * | * | | (+) The state of charge by the end of the flaying. | * | |
| Zahedi et al. [98] | | * | | * | | | * | | | SA, SEO, PSO | * | | | * | (1) (−) The starting time of visiting the suspected case with lowest priority by each ambulance and penalty time of visitation, (2) (−) the critical response time. | | * |
| Mamashli et al. [54] | | | * | | * | | * | | | HBMCGP-UF | * | | | * | (1) (−) Total time traveled of vehicles, (2) (−) the total environmental impacts of the system, (3) (−) the total demand's loss of all crisis points. | | * |

**Table 1.** *Cont.*

| Authors | Phase of Disaster | | | Model Type | | Problem Type | | | | Metaheuristic Algorithm | Time Period | | Objective Type | | Objective Function Minimize (−) Maximize (+) (the Number Indicates the Consecutive of the Objectives) | Results | |
|---|---|---|---|---|---|---|---|---|---|---|---|---|---|---|---|---|---|
| | Pre-Disaster | Post-Disaster | Integrated | Deter-ministic | Non-Deterministic | Facility Location | Distribution | Inventory | Mass Evacuation | | Single | Multi | Single | Bi/Multi | | Random Data | Case Study |
| Madani et al. [46] | | | * | * | | * | * | | * | NSGA-II, SA, VNS | * | | | * | (1) (+) the system reliability, (2) (−) the total cost of the relief logistic system. | * | |
| Khorsi et al. [47] | | | * | | * | * | | | | MOGF3EA | | * | | * | (1) (−) The arrival times of vehicles at the demand nodes during the planning, (2) (+) the reliability of the routes. | | * |
| Talebian Sharif and Salari [55] | | * | | * | | | * | | | GRASP | * | | * | | (−) The routing cost plus the allocation cost. | * | |
| Molladavoodi et al. [56] | | * | | | * | | * | | | GA | * | | | * | (1) (−) The total cost, (2) (−) the maximum unfulfilled demand. | | * |
| Akdoğan, et al. [70] | * | | | | * | * | | | | GA | * | | * | | (−) The frequency weighted mean response time of the system. | * | |
| Vahdani et al. [75] | | * | | * | | * | * | | | NSGA-II, MOPSO | | * | | * | (1) (−) The maximum vehicle route traveling time, (2) (−) the total cost. | * | |
| Huang and Song, [66] | | * | | | * | | * | | | CGA | * | | * | | (−) The total arrival time of the needed material. | * | |
| Babaei and Shahanaghi [67] | * | | | | * | * | | | | NSGA-II | * | | | * | (1) (−) The lost or logistics cost, (2) (+) demand satisfaction, (3) (+) the budget and the amount of demand response. | * | |

**Table 1.** *Cont.*

| Authors | Phase of Disaster | | | Model Type | | Problem Type | | | | Metaheuristic Algorithm | Time Period | | Objective Type | | Objective Function Minimize (−) Maximize (+) (the Number Indicates the Consecutive of the Objectives) | Results | |
|---|---|---|---|---|---|---|---|---|---|---|---|---|---|---|---|---|---|
| | Pre-Disaster | Post-Disaster | Integrated | Deterministic | Non-Deterministic | Facility Location | Distribution | Inventory | Mass Evacuation | | Single | Multi | Single | Bi/Multi | | Random Data | Case Study |
| Fathollahi-Fard et al. [101] | | | * | | * | | * | | | MICA, MWWO, MSSA, HWSA, HSIA | * | | * | | (1) (−) The cost of Hospital services and transportation. | * | * |
| Kim et al. [102] | | * | | | * | | * | | | AC | * | | | * | (1) (−) The weighted sum of total damages, (2) (−) competition time | * | |
| Sujaree and Samattapapong, [64] | | | * | * | | | * | | | HACROA | * | | * | | (−) Distance | * | |
| Shi et al. [81] | | | * | | * | | * | | | HGA | * | | * | | (−) Transportation cost | | * |
| Frifita et al. [63] | | | * | * | | | * | | | VNS | * | | * | | (1) (+) The number of visits assigned to each route, (2) (−) the traveling time. | | * |
| Decerle et al. [62] | | | * | * | | | * | | | HM-ACA | * | | * | | (−) The time needed to perform the care. | * | |
| Fathollahi-Fard [103] | | | * | * | | * | * | | | SA | * | | | * | (1) (−) The total cost of opening pharmacies and laboratories, (2) (−) environmental impact and green emissions. | * | |
| Saeidian et al. [104] | | | * | * | | * | | | | GA, BA | * | | * | | (−) The sum of all distances between centers and parcels. | * | * |
| Cao et al. [105] | | | * | * | | | * | | | GA | * | | | * | (1) (+) The lowest VPS (victims' perceived satisfaction) for all RDPs (relief demand points), (2) (−) the largest deviation on perceived satisfaction | | * |

**Table 1.** *Cont.*

| Authors | Phase of Disaster | | | Model Type | | Problem Type | | | | Metaheuristic Algorithm | Time Period | | Objective Type | | Objective Function Minimize (−) Maximize (+) (the Number Indicates the Consecutive of the Objectives) | Results | |
|---|---|---|---|---|---|---|---|---|---|---|---|---|---|---|---|---|---|
| | Pre-Disaster | Post-Disaster | Integrated | Deter-mini-stic | Non-Determi-nistic | Facility Location | Distri-bution | Inventory | Mass Evacua-tion | | Single | Multi | Single | Bi/Multi | | Random Data | Case Study |
| Qi and Hu [97] | | * | | * | | | * | | | ACS, PLS | | * | | * | (−) Total cost of distribution | * | |
| Su et al. [82] | | | * | * | | | * | | | SA | * | | * | | (−) Total travel time of disaster response coalitions and the total cost of the allocated emergency resources | * | |
| Zhang and Xiong, [106] | | * | | * | | | * | | | IACO | * | | | * | (1) (+) Demand satisfaction, (2) (−) total cost of grain distribution, (3) (−) distribution time | * | |
| Sharma et al. [83] | | | * | * | | * | * | | | TS | * | | * | | (−) Distance between hospitals and temporary blood centers | | * |
| Adarang et al. [69] | * | | | | * | | * | | | SFLA NSGA-II | * | | | * | (1) (−) Relief time, (2) (−) the total cost including location costs and the cost of route coverage by the vehicles | * | |
| Agarwal, Kant & Shankar [107] | | | * | * | | | * | * | * | PSA, GA | * | | | * | (1) (−) Total cost of facility establishment and drone procurement, (2) (−) The total number of uncovered customers | | * |
| Shavarani et al. [45] | * | | | | * | | * | | | NSGA-II, NSGA-III | | * | * | | (−) The total relief items supply chain cost | | * |
| Sadeghi moghadam and Ghasemian sahebi [89] | | | * | * | | | * | | | SA | * | | * | | (+) The demand coverage and reduce the rescue time | * | |

Table 1. *Cont.*

| Authors | Phase of Disaster | | | Model Type | | Problem Type | | | | Metaheuristic Algorithm | Time Period | | Objective Type | | Objective Function Minimize (−) Maximize (+) (the Number Indicates the Consecutive of the Objectives) | Results | |
|---|---|---|---|---|---|---|---|---|---|---|---|---|---|---|---|---|---|
| | Pre-Disaster | Post-Disaster | Integrated | Deterministic | Non-Deterministic | Facility Location | Distribution | Inventory | Mass Evacuation | | Single | Multi | Single | Bi/Multi | | Random Data | Case Study |
| Javadian et al. [108] | | | * | | * | * | * | | | NSGA-II, NRGA | * | | | * | (1) (−) The total operating cost of selected CWs and LDCs and inventory cost, (2) (−) the maximum travel time between each pair CW/LDC and demand point for the item | | * |
| Mosallanezhad et al. [87] | | * | | * | | | * | | | MOKA, MOSA, NSGA-II, MOKASA | | * | | * | (1) (−) Cost of the Personal Protection Equipment Supply Chain (2) (−) The amount of unsatisfied demands | | * |
| Buzón-Cantera et al. [90] | | * | | * | | | * | | | SA | | * | * | | (−) The penalty due to delays | * | |
| Korkou et al. [109] | | * | | * | | | * | | | DE, eDE, PSO, AP | | * | * | | (−) The shortages of different relief products | * | |
| Ferrer et al. [110] | | * | | | * | | * | | | RCA, ESCA, GRASP | * | | * | | (−) Total cost | | * |
| Hajipour et al. [58] | | | * | * | | | * | * | | MOVDO | * | | | * | (1) (−) The chain's total cost (2) (−) The number of undamaged items received by warehouses | * | |
| Ramezanian et al. [88] | | | * | | * | | * | | | MUCSOA | * | | | * | (1) (−) The total fuzzy transportation and inventory holding cost, (2) (−) unsatisfied demand, (3) (+) the minimum estimated demand ratios. | * | |

Table 1. *Cont.*

| Authors | Phase of Disaster | | | Model Type | | Problem Type | | | | Metaheuristic Algorithm | Time Period | | Objective Type | | Objective Function Minimize (−) Maximize (+) (the Number Indicates the Consecutive of the Objectives) | Results | |
|---|---|---|---|---|---|---|---|---|---|---|---|---|---|---|---|---|---|
| | Pre-Disaster | Post-Disaster | Integrated | Deter-ministic | Non-Deterministic | Facility Location | Distri-bution | Inventory | Mass Evacua-tion | | Single | Multi | Single | Bi/Multi | | Random Data | Case Study |
| Sadeghi et al. [89] | | * | | | * | * | | | | NSGA-II | * | | | * | (1) (−) The total cost of supplies shortage, (2) (−) the total cost of delivering supplies and the cost of constructing the distribution center, (3) (−) the total response time | | * |
| Caballero-Morales et al. [78] | * | | | * | | * | | | | KCM, GRASP-CKMC | * | | * | | (−) Total distance from each cluster to each assigned point | | * |
| Ransikarbum and Mason [96] | | * | | * | | | * | | | HNSGA-II | * | | * | | (−) Total cost of distribution | | * |
| Tofighi et al. [111] | | | * | | * | * | * | * | | DE | | * | * | | (−) Total operation cost of selected CWs | | * |
| Mardaninejad and Nastaran [71] | * | | | * | | * | | | | SA, PSO, ICA, ACO, ABC, FA, LAFA | * | | * | | (−) The distance and fixed cost of equipping a temporary accommodation center | | * |
| Foroughi et al. [112] | | * | | | * | | * | | | LP-GA | * | | | * | (1) (−) Total cost, (2) (+) each facility's weighted resilience levels | | * |
| Golabi et al. [85] | * | | | | * | * | | | | GA, MA, SA | * | | * | | (−) The aggregate traveling time | | * |
| Nayeri et al. [72] | * | | * | | * | * | * | | | SA, PSO, SA-PSO | * | | * | | (1) (−) the sum of the weighted completion time of the relief operation | | * |
| Dávila de León et al. [93] | | * | | * | | | * | | | SA, GRA | * | | * | | (−) The time required to provide humanitarian aid | * | |

**Table 1.** *Cont.*

| Authors | Phase of Disaster | | | Model Type | | Problem Type | | | | Metahe-uristic Algorithm | Time Period | | Objective Type | | Objective Function Minimize (−) Maximize (+) (the Number Indicates the Consecutive of the Objectives) | Results | |
|---|---|---|---|---|---|---|---|---|---|---|---|---|---|---|---|---|---|
| | Pre-Disaster | Post-Disaster | Integrated | Deter-mini-stic | Non-Determi-nistic | Facility Location | Distri-bution | Inventory | Mass Evacua-tion | | Single | Multi | Single | Bi/Multi | | Random Data | Case Study |
| Hasani and Mokhtari [73] | * | | | | * | * | | * | | NSGA-II, PSO | * | | | * | (1) (+) The total coverage by the relief network, (2) (−) the total cost, (3) (−) the maximum risk of total demand nodes | | * |
| Zhu et al. [113] | | * | | * | | | | | * | ACO | * | | | * | (1) (−) The transportation cost, (2) (−) the absolute deprivation cost, (3) (−) relative deprivation cost | | * |
| Danesh et al. [114] | | * | | | * | | * | | | GOA | * | | | * | (1) (+) The total value made by evaluating the sites and roads, (2) (+) the minimum cover of sites, (3) (+) the minimum cover of roads | | * |
| Hoseininezhad et al. [115] | * | | | | * | * | | | | NSGA II | * | | | * | (1) (−) The transportation cost of injured people, (2) (+) the impact of factor k on the location of relief chain h, (3)(−) the time of transferring injured people, (4) (−) the deviation of the capacity | | * |
| Edrisi et al. [94] | * | | | * | | | * | | | PSO | * | | * | | (−) death toll | * | |
| Torabi et al. [116] | | | * | | * | * | | | | DE | | * | * | | (−) total cost | | * |

**Table 1.** *Cont.*

| Authors | Phase of Disaster | | | Model Type | | Problem Type | | | | Metaheuristic Algorithm | Time Period | | Objective Type | | Objective Function Minimize (−) Maximize (+) (the Number Indicates the Consecutive of the Objectives) | Results | |
|---|---|---|---|---|---|---|---|---|---|---|---|---|---|---|---|---|---|
| | Pre-Disaster | Post-Disaster | Integrated | Deter-ministic | Non-Deterministic | Facility Location | Distribution | Inventory | Mass Evacuation | | Single | Multi | Single | Bi/Multi | | Random Data | Case Study |
| Ghasemi et al. [117] | | | * | | * | * | * | | | NSGA II | | * | | * | (1) (−) The number of injured people who a not serviced, (2) (−) the cost of relief supplies | | * |
| Hu et al. [118] | | * | | | * | | * | | | PSO | * | | | * | (1) (+) The overall utility the relief resources to achieve the efficiency purpose, (2) (+) the minimal satisfaction rate | | * |
| Nayeri et al. [119] | * | | | * | | | | | * | GA, PSO | * | | | * | (1) (−) the sum of the weighted completion time of the relief operation, (2) (−) the sum of deprivation times | * | |
| Wang et al. [120] | | * | | | * | | * | | | MOCGA | | * | | * | (1) (−) Disaster losses, (2) (−) transportation risks | * | |
| Xu et al. [121] | | * | | * | | | * | | | PSO | * | | * | | (−) Cost of rescue plan | | * |
| Sabouhi and Bozorgi-Amiri [86] | | * | | * | | | * | | | MA | * | | * | | (−) The total waiting time of evacuees and delivery time of supplies | | * |
| Wex et al. [92] | | * | | * | | | * | | | GRASP | * | | * | | (−) The sum of completion times | * | |

## 6. Main Findings and Conclusions

In a world where there are more natural and man-made disasters, and massive diseases that require the attention of HSC, it is necessary to consider what types of problems are being addressed, their classification and the types of metaheuristics used to solve them. In this work, 80 articles were selected considering the systematic literature review methodology, aiming to answer two research questions: RQ1 How are the HSC problems that have been solved from metaheuristics since 2015 classified? and RQ2 What is the gap found to accomplish future research in metaheuristics in HSC?

After reviewing all the selected articles, it was found that the deterministic and non-deterministic problems are well balanced with 52.56% of the deterministic in contrast to 47.44% of the non-deterministic. The highest number of addressed problems were regarding distribution with 71.79%, followed by facility location with 41.03%, it should be noted that there are research papers that consider both problems together. In contrast, the least resolved problems are inventory with 11.54% and Mass Evacuation with 10.26%. In the analysis, it is also concluded that the post-disaster phase is the most considered one with 51.28% followed by the one that integrates both phases, pre- and post-disaster with 34.62%, and the pre-disaster phase with 14.10% (RQ1). In this regard, there are areas of opportunity to address the problem of inventory and mass evacuation considering the pre-disaster or both integrated phases. Consequently, the most convenient thing is considering problems in a non-deterministic way to deal with the uncertainty that exists in the current context (RQ2).

In the case of single or multiple objectives, they are balanced, with 53.85% being multiple objectives, while 46.15% of the problems are single. Regarding the period, single-period problems are further solved, with 76.92% against 23.07% of multiple-period problems (RQ1). This may be due to the complexity that multi-period problems represent, however, solving problems with multiple periods would allow that the proposed models are closer to reality (RQ2).

As to MHs, it was found that 71.79% were based on metaphor; of these 23.07% of the total articles are evolutionary, the most common being Genetic Algorithms. In the Swarm Intelligence area, there is 30.76% of the total work, with Particle Swarm Optimization being the one that is the most extensively occupied. Based on physics there are 15.38% of articles with Simulated Annealing. From those not based on metaphors, there are a total of 28.20%, the most widely used being the search algorithm with 15.38%. Of the MHS that solve multiple objectives, there are 38.46% of the articles in which the NSGA-II predominates with 15 solved papers. Using hybrid algorithms there is 15.38% (RQ2).

After analyzing the articles, what they have in common is the modeling of the objective function and some solution methods; in contrast, they adapt the restrictions to the type of problem to be solved. When reviewing deeply, some articles that have been written by the same author have adapted his original proposal to solve other more extensive problems or with other products and movements in HSC.

The future research could be in two approaches; (1) Test the new MHs that are mentioned in Section 3 in already structured problems concerning the four main problems—facility location, distribution, inventory, mass evacuation—and do serious statistical analysis to see which one converges faster to the solution. When authors are applying new MHs to solve problems related to HSC, researchers should statistically compare different MHs to decide which one is the best for the problem that they are trying to solve. Some works present the comparison of different solution methods through statistic techniques, for example, Page trend test for convergence analysis [122] or Carrasco et al. [123] who survey current trends in statistical analysis proposals for the comparison of computational intelligence algorithms.

(2) Use the proposed modeling to solve real cases, not only at the proposal level of the modeling or testing the MHs, but also to impact the community. When faced with this type of problem such as the distribution of vaccines, which has a global scale, it will be necessary to make use of multi-objective problems, which are solved in several phases, or even

generate diverse solutions with high performance such as Quality-Diversity optimization. What is observed in the articles is that the cases are made to test one model or different MHs, but they do not apply to real problems that are highly complex.

Considering the complexity of the multi-period and multi-objective models that must be solved in real HSC situations, multi-objective MHs such as the NSGA-II play an extremely important role, which is why they will continue to be used to solve increasingly complex problems. There is a very important area of opportunity in solving these problems related to HSC using new proposals of multi-objective MHs; seemingly, they could also be hybridized with other heuristics and MHs to generate convergence for good solutions in a reasonable computational time (RQ2).

In summary, we point out that providing the relevance and complexity of these types of problems, future research in HSC should be done in non-deterministic and multi-period problems, which integrate pre- and post-disaster stages, and that gradually include problems such as inventory and mass evacuation and in which new multi-objective MHs (RQ1 and RQ2) are tested.

**Author Contributions:** Conceptualization, E.S.H.-G. and F.S.R.; methodology, E.S.H.-G. and F.S.R.; validation, N.H.-G. and R.G.M.; formal analysis, E.S.H.-G.; investigation, E.S.H.-G. and F.S.R.; resources, N.H.-G.; writing—original draft preparation, E.S.H.-G. and F.S.R.; writing—review and editing, E.S.H.-G. and F.S.R.; visualization, F.S.R.; supervision, E.S.H.-G.; project administration, N.H.-G. All authors have read and agreed to the published version of the manuscript.

**Funding:** This research received no external funding.

**Data Availability Statement:** Data is contained within the article.

**Conflicts of Interest:** The authors declare no conflict of interest.

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
