# Peer review of "Metaheuristics in the Humanitarian Supply Chain"

_algorithms, doi:10.3390/a14120364_

Round 1

Reviewer 1 Report

All acronyms must be defined including GRASP.

The contributions of the article must be emphasized in terms of originality, significance, and performance metrics in the abstract and introduction.

Abstract should include the significance and importance of the work in terms of metaheuristics.

Recently-proposed metaheuristics (in general terms) from high impact factor journal (see https://www.scimagojr.com/) should be cited like from IEEE transactions, Springer and Elsevier in the introduction or in a related work section or related to Figs. 7 and 8.

Examples:

https://ieeexplore.ieee.org/document/6819057

https://ieeexplore.ieee.org/document/8721125

https://www.sciencedirect.com/science/article/abs/pii/S0957417420305224

https://www.sciencedirect.com/science/article/abs/pii/S1359431119301644

https://www.sciencedirect.com/science/article/abs/pii/S0952197619302167

https://www.sciencedirect.com/science/article/pii/S0952197619302775

https://www.sciencedirect.com/science/article/pii/S1568494619301309

https://www.sciencedirect.com/science/article/abs/pii/S2451904919300162

https://www.sciencedirect.com/science/article/pii/S0952197619302593

https://www.sciencedirect.com/science/article/abs/pii/S1568494619308002

Comments about statistical analysis of metaheuristics results could be presented.

See examples in the following papers:

Recent trends in the use of statistical tests for comparing swarm and evolutionary computing algorithms: Practical guidelines and a critical review

Swarm and Evolutionary Computation Volume 54 May 2020Article 100665

  1. Carrasco, S. García, M. M. Rueda, S. Das, F. Herrera

https://www.sciencedirect.com/science/article/pii/S2210650219302639

Analyzing convergence performance of evolutionary algorithms: A statistical approach

Information Sciences Volume 28924 December 2014 Pages 41-58

Joaquín Derrac, Salvador García, Sheldon Hui, Ponnuthurai Nagaratnam Suganthan, Francisco Herrera

https://www.sciencedirect.com/science/article/pii/S0020025514006276

Fig. 6 is with low resolution.

Equations could be presented in Section 5.2 (Time period, objective type and objective function).

Author Response

Dear Reviewer,

In the next lines are explained your comments and how we attend them one by one.  Thank you for your suggestions, they contribute a lot for us.

  1. All acronyms must be defined including GRASP.

This was attended in lines 198-291.

  1. The contributions of the article must be emphasized in terms of originality, significance, and performance metrics in the abstract and introduction.

Regarding the originality of the article, to our knowledge there has not been a review article on metaheuristics applied to HSC, and knowing the types of problems that have been solved and the most used MHs can help researchers to know what is missing to do and help to solve real situations. This explanation was included in the abstract and in the introduction in lines 102-105.

  1. Abstract should include the significance and importance of the work in terms of metaheuristics.

In real situations where the problems grow computationally, the metaheuristics are the indicated solution algorithms and all cases of HSC, generally, are problems with these characteristics, this as included in the abstract.

  1. Recently-proposed metaheuristics (in general terms) from high impact factor journal (see https://www.scimagojr.com/) should be cited like from IEEE transactions, Springer and Elsevier in the introduction or in a related work section or related to Figs. 7 and 8.

Examples:

https://ieeexplore.ieee.org/document/6819057

https://ieeexplore.ieee.org/document/8721125

https://www.sciencedirect.com/science/article/abs/pii/S0957417420305224

https://www.sciencedirect.com/science/article/abs/pii/S1359431119301644

https://www.sciencedirect.com/science/article/abs/pii/S0952197619302167

https://www.sciencedirect.com/science/article/pii/S0952197619302775

https://www.sciencedirect.com/science/article/pii/S1568494619301309

https://www.sciencedirect.com/science/article/abs/pii/S2451904919300162

https://www.sciencedirect.com/science/article/pii/S0952197619302593

https://www.sciencedirect.com/science/article/abs/pii/S1568494619308002

 We include a review of the proposed metaheuristics in lines 296-370. We also modify figure 5 on line 458.

  1. Comments about statistical analysis of metaheuristics results could be presented.

See examples in the following papers:

Recent trends in the use of statistical tests for comparing swarm and evolutionary computing algorithms: Practical guidelines and a critical review

Swarm and Evolutionary Computation Volume 54 May 2020Article 100665

  1. Carrasco, S. García, M. M. Rueda, S. Das, F. Herrera

https://www.sciencedirect.com/science/article/pii/S2210650219302639

Analyzing convergence performance of evolutionary algorithms: A statistical approach

Information Sciences Volume 28924 December 2014 Pages 41-58

Joaquín Derrac, Salvador García, Sheldon Hui, Ponnuthurai Nagaratnam Suganthan, Francisco Herrera

https://www.sciencedirect.com/science/article/pii/S0020025514006276

Dear reviewer, in his article Derrac uses the Page's test to compare different algorithms to solve the same continuous function, Carrasco et. al (2020) proposes different statistical analyzes to test the convergence of the algorithms but in the same function. The models used in HSC these are multiobjective and subject to a set of restrictions, including some of them made specifically for a particular solution so I don't think we could compare them through these techniques, maybe just issue a little comment for the researchers that test different MHs in their problems , lines 1080-1084.

  1. 6 is with low resolution.

This has already been addressed. 

7-Equations could be presented in Section 5.2 (Time period, objective type and objective function).

Dear reviewer, as you can see in lines 763 to 823, the objective functions are so diverse that it is difficult to place the equations of the objective functions, in fact, many works are multiobjective. The review articles that we review simply classify them but do not put any equation due to the complexity they represent, even if we try to do it by type of problem, many of them have two or more proposed objectives. The review articles are

  • Habib, M. S.; Lee, Y. H.; Memon, M. S. Mathematical Models in Humanitarian Supply Chain Management: A Systematic Literature Review. Math. Probl. Eng. 2015, Vol. 2016, 1- 20.
  • Manopiniwes, W.; Irohara, T. A Review of Relief Supply Chain Optimization. J. Humanit. Logist. Supply Chain Manag. 2014, Vol. 13, 1 – 14. 

Best regards, 

Eva

Reviewer 2 Report

The paper comprises a systematic review of metaheuristics applied to humanitarian supply chain problems. The survey approach seems sound, and a comprehensive summary is given of work in this area. However, there are a few ways the paper should be improved.

1. Line 177: "The most commonly used metaheuristics for solving problems..." The ordering of algorithms in the following section is a bit odd. It's not alphabetical or by popularity (though the latter is hard to measure; ) Scatter search before GAs? 

2. Line 184: "GRASP and Path Rethinking." pleas provide citations.

3. Section 5.1 covers both abstract problem definitions (facility loc/distribution/inventory/evacuation), and characteristics of problems ((non)deterministic/pre/post disaster). Given that 5.2 covers another characteristic, single/multi objective, it might be worth separating 5.1 into two parts.

4. My biggest criticism: the conclusions and future work are rather brief, and descriptive rather than providing much analysis. Is there anything in common across HSC problems that would allow solutions in one problem area to carry over to another? Is there scope for applying the newest the newest developments in methaheuristics? Many-objective algorithms, quality-diversity algorithms, large-scale global optimisation, surrogate models, matheuristics? Some speculation here, and analysis of whether any of these are already finding applications in HSC problems would be most welcome.

5. Minor: On the whole the paper is well written, but there are some slightly odd expressions - please proof read to ensure readability. A couple of examples:
Line 18: "it was obtained that future..." > "it s concluded that future..."
Line 56 "The other aspect that HSCs point out" > "The other aspect that HSCs raise"
Line 115 "it is noted prominent"  - delete either noted or prominent

Author Response

Reviewer 2:

Dear Reviewer,

In the next lines are explained your comments and how we attend them one by one.  Thank you for your suggestions, they contribute a lot for us.

The paper comprises a systematic review of metaheuristics applied to humanitarian supply chain problems. The survey approach seems sound, and a comprehensive summary is given of work in this area. However, there are a few ways the paper should be improved.

  1. Line 177: "The most commonly used metaheuristics for solving problems..." The ordering of algorithms in the following section is a bit odd. It's not alphabetical or by popularity (though the latter is hard to measure; ) Scatter search before Gas.

Thank you for your suggestions, this was  attended an now is in alphabetical order in lines 299-370, and also, other reviewer suggests us to incorporate recently proposed  MHs thtar are also incorporated in lines  299-370.

  1. Line 184: "GRASP and Path Rethinking." pleas provide citations.

This was attended.

  1. Section 5.1 covers both abstract problem definitions (facility loc/distribution/inventory/evacuation), and characteristics of problems ((non)deterministic/pre/post disaster). Given that 5.2 covers another characteristic, single/multi objective, it might be worth separating 5.1 into two parts.

We separate 5.1 in two sections as you suggested.

  1. 4. My biggest criticism: the conclusions and future work are rather brief, and descriptive rather than providing much analysis.

Is there anything in common across HSC problems that would allow solutions in one problem area to carry over to another?

After analyzing the articles, what they have in common is the modeling of the objective function and some solution methods, in contrast they adapt the restrictions to the type of problem to be solved. When reviewing deeply, some articles that have been written by the same author have adapted his original proposal to solve other more extensive problems or with other products and movements in HSC, this has already been added in the lines 1069-1073,

Is there scope for applying the newest developments in methaheuristics? Many-objective algorithms, quality-diversity algorithms, , surrogate models, matheuristics? Some speculation here, and analysis of whether any of these are already finding applications in HSC problems would be most welcome.

The future research could be in two approaches; 1) Test new MHs , that are mention in section 2, in already structured problems with respect to the four main problems such as facility location, distribution, inventory, mass evactuation and do serious statistical analysis to see which one converges faster to the solution, lines(1074-1077)

2) Use the proposed modeling to solve real cases, not only at the proposal level of the modeling or test the MHs, but also to impact the community. When faced with this type of problem such as the distribution of vaccines, which has a global scale, it will be necessary to make use of multiobjective problems, which are solved in several phases, or even generate diverse solutions with high performance such as Quality-Diversity optimization,  what is observed in the articles is that the cases are made to test one model or different MHs, but they do not apply to real problems that are highly complex, (lines 1085-1091)

  1. Minor: On the whole the paper is well written, but there are some slightly odd expressions - please proof read to ensure readability. A couple of examples:
    Line 18: "it was obtained that future..." > "it s concluded that future..."
    Line 56 "The other aspect that HSCs point out" > "The other aspect that HSCs raise"
    Line 115 "it is noted prominent"  - delete either noted or prominent.

We use a service for proof read and these details were corrected int the paper.

Best regards, Eva

Round 2

Reviewer 1 Report

The current version of the paper is improved.

Figs. 1 and 2 are with low resolution. Please verify it.

The article is organized as follows, in section 2, the definition

->

The remainder of this article is organized as follows, in section 2, the definition

Reviewer 2 Report

I think my concerns have now been addressed.